# A Mediterranean drifters' dataset

Alberto Ribotti[1], Antonio Bussani[2], Milena Menna[2], Andrea Satta[1], Roberto Sorgente[1], Andrea Cucco[1], Riccardo Gerin[2]

[1]Istituto per lo studio degli impatti Antropici e Sostenibilità in ambiente marino (IAS) of CNR, 09170 Oristano, Italy, https://orcid.org/0000-0002-6709-1600,https://orcid.org/0000-0001-8411-1872, https://orcid.org/0000-0003-0268-7822, https://orcid.org/0000-0002-4469-2286
[2]Istituto Nazionale di Oceanografia e di Geofisica Sperimentale-OGS, Borgo Grotta Gigante, 42/c - 34010 Sgonico (Trieste), Italy, https://orcid.org/0000-0003-0340-3078, https://orcid.org/0000-0002-0149-0502, https://orcid.org/0000-0002-9788-0803

*Correspondence to*: Alberto Ribotti (alberto.ribotti@cnr.it)

**Abstract.** Over a hundred experiments were conducted between 1998 and 2022 in the Mediterranean Sea using surface Lagrangian drifters, at coastal and offshore level. Raw data was initially unified and pre-processed manually by eliminating spikes and wrong positions or date/time information. The integrity of the received data packages was checked, and incomplete ones were discarded. Deployment information was retrieved for each drifter and integrated into the PostgreSQL database, realised, and maintained by the National Institute of Oceanography and Applied Geophysics (OGS) in Trieste (IT). This database also collects a variety of metadata about the drifter model, project, owner, and operator. Subsequently data were processed using standard procedures of editing and quality control developed for the OGS drifter dataset to remove spikes generated by malfunctioning of the sensors and obtain files with common characteristics. Drifter data and plots of each track were also visually checked to remove any point not identified by the automatic procedure and clearly erroneous. Drifters' trajectories were split into two or more segments that have been considered as different deployments, in case of specific drifters' behaviours. Data were interpolated at defined time intervals obtaining a dataset of 158 trajectories, available from the public open-access repository in SEA scieNtific Open data Edition (SEANOE) at https://doi.org/10.17882/90537 (Ribotti et al., 2022), in two version: one compliant to the Copernicus format and the other usable with Panoply netCDF viewer , and in SeaDataNet at https://cdi.seadatanet.org/search/welcome.php?query=2610&query_code={9F00DF80-1881-42DD-9DF1-B9BD0282F2B0}.

**Keywords:** Mediterranean, drifter, Lagrangian data, surface circulation, quality control

## 1 Introduction

In oceanographic research since the early 1980s, extensive use has been made of surface drifters to study ocean surface dynamics, particularly during the U.S. Coastal Ocean Dynamics Experiment (CODE) described by Davis (1985), with the design, testing and use of light weight, inexpensive drifters. They were tracked by radio direction finding triangulation and also the new satellite Global Positioning System (GPS) launched in 1978. These drifters, named CODE, are still used today, greatly improved in their data transmission systems.

In general, drifters are designed to follow the sea currents for long distances while minimising the direct effects of wind and waves acting on the elements protruding outside the sea surface.

In 1991 the Global Ocean Observing System (GOOS) programme started, led by the Intergovernmental Oceanographic Commission (IOC) of UNESCO followed, in 1994, by its European component EuroGOOS that highlighted the operational oceanography value for society (Woods et al., 1996). The activities related to operational oceanography promoted the use of drifters also for the management of emergencies at sea, like oil spills or contaminants (Pisano et al., 2016), mitigation of extreme events (Goni et al., 2017; Menna et al., 2023), and the validation of numerical forecasting systems (De Dominicis et al., 2016; Sorgente et al., 2016).

The Italian National Research Council in Oristano (CNR hereafter), uses drifters for research purposes linked with scientific projects, mainly focused on the study of local or sub-basin surface dynamics or on the calibration and validation of oceanographic prediction systems, in the framework of physical and operational oceanography.

CNR started its activities with drifters in 1998-1999. Early activities consisted in the usage of a single drifter in 15 coastal experiments for six months, along with the use of a multiparametric probe, to study the hydrodynamics of the Gulf of Oristano (Table 1), western Sardinia. The adopted instrument was a Coastal Lagrangian Drifter (CLD) designed and realised by a small Italian enterprise equipped with GPS and digital network (GSM) data transmission (Ribotti et al., 2000, 2002). Due to technical problems, experiments were interrupted but restarted ten years later with different objectives and the adoption of a different type of drifter. In 2009 and 2010, CNR implemented a numerical oceanographic and oil spill prediction system limited to the Bonifacio Strait area in collaboration with the local Coast Guard. For the calibration and validation of the implemented numerical models, 9 experiments (Table 1) were conducted inside and outside the Bonifacio Strait by using US CODE drifters with satellite transmission (Cucco et al., 2012; Ribotti et al., 2013). As some experiments were carried out in La Maddalena Archipelago, a coastal area characterised by narrow channels and small islands, due to the high risk of stranding, CNR modified the instruments inserting a switch, to turn them on or off, useful to re-use the recovered drifters.

In the framework of operational oceanography, in September 2014 CNR participated in an international exercise at sea on oil spill combat and Save And Rescue (SAR) activities launching three new Spanish satellite drifters, named Ocean Drifter (oDi; Table 1), with solar panel and temperature sensor, specifically designed for oil spill studies. After the exercise, drifters were released in western and southern Sardinian coastal waters to investigate the main surface hydrodynamics.

From the end of 2015 onwards, new GPS, cost effective, handy, and durable drifters produced by a Spanish company, were adopted by CNR in several field activities. Different types of instruments were used, feasible for coastal (with GPRS transmission) or for offshore areas (with satellite transmission), with a switch and rechargeable batteries that permitted the use of the same drifter in different experiments. GPRS (or General Packet Radio Service) was the first cellular system specifically designed for packet-switched, medium-speed data transfer over a cellular network, so it can only be used in coastal areas covered by a cellular network and it is less expensive than GPS. These drifters were deployed in experiments all over the central Mediterranean Sea (Table 1) with data acquisitions ranging from few hours to over 12 months for purposes linked to both physical/biological (Quattrocchi et al., 2021a, b) or operational oceanography activities (Ribotti et al., 2019; Sorgente et al., 2020).

Recently, the OGS in Trieste has re-processed all drifters' experiments following standard and state-of-the-art procedures (editing and interpolation) already adopted for previously released Lagrangian datasets, then creating a new one freely available online in two formats: Copernicus and NASA/NOAA-like (manageable by Panoply, a NASA-developed data viewer).

In this paper we describe the drifters' characteristics, the procedures of data acquisition and processing in detail.

## 2. Drifters

The CNR conducted over 138 experiments in the Mediterranean basin with surface Lagrangian drifters in 12 years, not continuously, between July 1998 and April 2022 (month of the last recovery), at coastal and offshore level (Table 1 and Fig. 1).

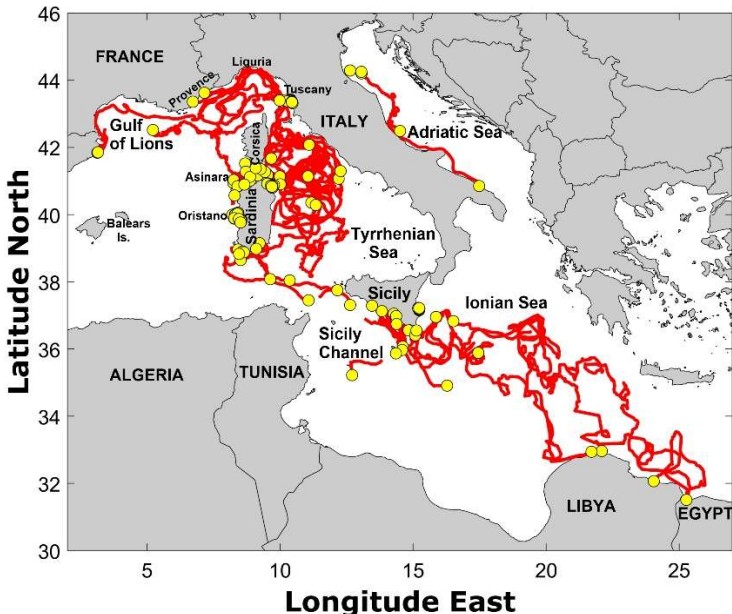

80
81  Figure 1. In red all drifters' trajectories acquired during the experiments between 1998 and 2022 (yellow dots represent
82  the position of the deployment).

83

| Year | Start Month | # Experiments | Start Area | Type of drifter |
|---|---|---|---|---|
| 1998 | July | 1 | Oristano Gulf | CLD |
| | Aug. | 6 | Oristano Gulf | CLD |
| | Oct. | 3 | Oristano Gulf | CLD |
| 1999 | Jan. | 5 | Oristano Gulf | CLD |
| 2009 | May | 1 | Asinara Gulf | CODE |
| | | 1 | Tyrrhenian Sea | CODE |
| | June | 2 | Bonifacio strait | CODE |
| | Aug. | 1 | Bonifacio strait | CODE |
| 2010 | Mar. | 2 | Bonifacio strait | CODE |
| | Sept. | 1 | Bonifacio strait | CODE |
| | | 1 | Tyrrhenian Sea | CODE |
| 2014 | Sept. | 1 | South Sardinia | ODi |
| | Oct. | 1 | West Sardinia | ODi |
| 2015 | Dec. | 5 | North Tyrrhenian | LCA |
| 2016 | Feb. | 5 | North Tyrrhenian | LCA |
| | March | 4 | North Tyrrhenian | LCA |
| | July | 1 | Cagliari Gulf | LCA |
| 2017 | March | 6 | West Sardinia | LCA |
| | June | 4 | West Sardinia | LCA |
| | | 3 | Sicily | LCA |
| | July | 1 | Sicily | LCA |
| | Oct. | 14 | Sicily | LCA, LCE |
| | Nov. | 4 | Sicily | LCE |
| 2018 | May | 4 | North Adriatic | LCA, LCE |

| | | 2 | Sicily Channel | LCE |
|---|---|---|---|---|
| | June | 1 | South Adriatic | LCE |
| | | 1 | West Sardina | LCA |
| | July | 1 | West Sardina | LCA |
| | | 3 | N-E Sardinia | LCA |
| | Sept. | 3 | Tyrrhenian Sea | LCE, LCH |
| | | 10 | Asinara Gulf | LCA, LCE |
| | | 1 | Gulf of Lions | LCE |
| 2019 | June | 1 | North Adriatic | LCE |
| | | 2 | N-E Sardinia | LCA |
| | July | 2 | N-E Sardinia | LCA |
| | Sept. | 6 | Asinara Gulf | LCA, LCE |
| | | 4 | N-E Sardinia | LCA |
| | Oct. | 1 | West Sardinia | LCE |
| | Nov. | 4 | N-E Sardinia | LCA |
| 2020 | May | 2 | Port of Olbia | LCA |
| | Oct. | 9 | Asinara Gulf | LCA, LCE, LCH |
| 2021 | Oct. | 2 | South Sardinia | LCE |
| | | 1 | Tyrrhenian Sea | LCE |
| | Nov. | 5 | South Sardinia | LCE, LCF |

84

Table 1. List of the 138 experiments between 1998 and early 2022. Acronyms indicate drifters per type: CLD, CODE, ODi, and the SouthTEK Nomad family LCA (GPRS), LCE (offshore), LCH (hybrid), LCF (with temperature sensor). Dates (year and month) and Start Area indicate when/where the drifter was initially deployed.

Lagrangian drifters produced by 4 different companies have been used in these years, with different characteristics in data transmission, structure, repeatability of the experiments, dimensions, batteries, management of the experiments.

**2.1 Tracks 1998-1999: Coastal Lagrangian Drifter or CLD**

The CLD was produced by the Italian company InnoTech S.c.r.l. and designed just for coastal use. It transmitted its GPS position, by a Trimble Lassen™ SK8, at a frequency of 5 minutes by a GSM mobile phone. The maximum operating time of the buoy was approximately 72 hours. The housing of the drifting buoy was in PVC with an electronic unit, a rechargeable battery pack and antennas. Dimensions and weight were 140 cm high (h) x 27 cm in diameter (d) and 12.5 Kg, respectively (Fig. 2A). A sail (0.5 m length and diameter) was attached below the drifter to enhance the drag below the water surface. The acquired position data was transmitted through a commercial modem to dedicated software on a computer. This software, in a Windows™ environment, allowed the automatic reception of data from the buoy, provided for the control of the correct functioning of the system and for a quick and easy setting of the operating parameters (selection of the buoys used, interval of acquisition of the data, etc.). Transmitted data were collected into files in several formats including ASCII format with the extension DAT. This drifter was used for about six months, between July 1998 and January 1999 (Table 1), for experiments of a few hours to study the surface circulation of the Gulf of Oristano (western Sardinia).

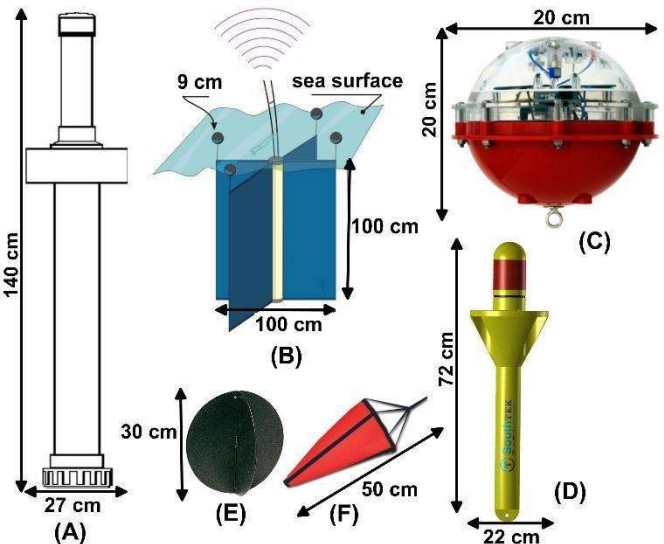


Figure 2. The four types of drifters used with their dimension in centimetres: A) CLD; B) CODE; C) ODi; D) LC; E) Pila
drogue; F) Satis drogue (credits: ODi (C) from Albatros' leaflet; LC (D) and drogues (E; F) from SouthTek's website)

## 2.2 Tracks 2009-2010: CODE drifters

Between May 2009 and September 2010 (Table 1), CNR used the ArgoDrifter or CODE by Technocean (FL, USA) for
studies in northern Sardinia. The instrument dimensions were 100 cm (h) x 100 cm (d) (Fig. 2B) and consisted of a
cylinder containing batteries and electronics and four arms placed at 90° each other, supporting four sails, for a total area
of about 2 m$^2$. Batteries permitted operation of a year with an hourly data acquisition frequency. CODE drifters were
fitted with an ARGOS satellite transmitter, a GPS, and a temperature sensor. Drifter position was measured by satellite
both ARGOS and GPS. Operatively GPS and ARGOS differ substantially in their accuracy of the positioning
measurements. GPS accuracy has an average error of 4 m, with an ellipse of variance of axes of about 5-7 metres (Barbanti
et al., 2005); the position measured by ARGOS varies being linked with the number of visible satellites used daily from
a minimum of 1 with an error of about 1.5 km to 4 or more satellites with less than 50 m of error. Direct slip measurements
(Poulain et al., 2002; Poulain and Gerin, 2019), with acoustic current metres, show that CODE drifters follow surface
currents with a tolerance of 0.1 percent of the wind speed and a movement consistent with the Ekman dynamics near the
surface and a velocity component to the right of the prevailing wind. The wind-induced slips and the Ekman surface
currents can also be estimated from drifter data using simple regression models which include complex drifter velocities
and surface wind products (Ralph and Niiler, 1999; Rio and Hernandez, 2003; Centurioni et al., 2009; Poulain et al, 2009,
2012, 2013). These models show that the CODE wind-driven currents (slip + Ekman + Stokes) in the Mediterranean are
about 1% of the wind speed, at an angle of about 30° to the right of the wind.
Drifters were set to measure their position every 4 minutes during each experiment. In 2010, CNR modified CODE drifters
inserting an external on/off switch, not present in the original instrument. This made it possible to carry out different
experiments with the same instrument even after months. Data was downloaded from the ArgosWeb site, managed by the
French Collecte Localization Satellites (CLS), in ASCII and/or in binary format. Subsequently they were subjected to
post-processing, using Matlab codes provided by the OGS in Trieste. The median of the data was calculated for each
interval then eliminating data outside the range established by the mean +/- three times their standard deviation.
This type of drifters was mainly used in northern Sardinia (Asinara Gulf and Bonifacio Strait) with some trajectories
acquired also in the northern Tyrrhenian Sea. Experiments have ranged from a few hours to over one month with the aim
of studying the circulation in the Bonifacio Strait and La Maddalena Archipelago and to validate a forecasting system for
oil spill combat (Cucco et al., 2012; Ribotti et al., 2013) in the framework of the Italian SOS Bonifacio project (Ribotti et
al., 2013).

## 2.3 Tracks 2014: Iridium Ocean Drifter (ODi)

In September-October 2014 (Table 1), CNR used the Iridium Ocean Drifter (ODi), made by the Spanish Albatros Marine
Technology SA. It was a small, low-cost, and compact surface buoy to track sea currents by a GPS module and transmits
data via Iridium satellite system (Short Burst Data - SBD), a global full ocean coverage bidirectional satellite
communication network. It was composed of two identical halves of a spherical drifter of 20 cm in diameter (Fig. 2C)
and about half of it protruded above the sea surface. The ratio of drag area in the water to drag area outside the water was
16.9 (Callies et al., 2017). This makes it optimal for oil spill tracking and search and rescue operations. Its 5-litre volume
and 3 Kg of weight allowed the use of a holey-sock drogue, while the presence of a solar power charging module, realised
to reduce battery size, gave a theoretically unlimited autonomy. Standard measurements were GPS position/time,
temperature, and battery level. The sampling frequency and transmission frequency were user-configurable through its
software and internet connection. A sail, similar to that described for CLD drifter, was attached below every drifter. Data
was acquired with a frequency of 20-30 minutes. Despite the interesting structure suitable for studies on oil spills at sea,
the drifter showed some technical problems that limited its use in long experiments. A first launch was scheduled in
September 2014 in the Gulf of Cagliari, south Sardinia, with an acquisition over one month long in the framework of an
international exercise at sea, named Squalo2014, coordinated by the local Coast Guard. Data was used to validate a high-
resolution ocean oil-spill forecasting model (Sorgente et al., 2015). Another short deployment, of less than 6 hours, was
made a few nautical miles off the Oristano Gulf, western Sardinia.
**2.4 Tracks 2015-2022: coastal and offshore Nomad drifters**
Since December 2015 (Table 1), CNR has been using Lagrangian drifters of the Nomad family produced by the Spanish
SouthTEK Sensing Technologies S.L. and used by the scientific community (Bolado-Penagos et al., 2020; Sala et al.,
2022). The buoys are of three types: coastal GPRS, offshore satellite and hybrid, which can use either GPRS under mobile
coverage or satellite transmission. Both GPRS drifters, namely the Coastal Nomad, and the satellite ones, the Offshore
Nomad, are made of plastic, yellow colour, 72 cm (h) x 22 cm (d) (Fig. 2D) with a weight of 2.895 Kg. The Hybrid
Nomad drifters are the same. The lithium batteries allow operations up to 7 days to the GPRS and several months to the
satellite drifters. When in the water, only the yellow cylindrical head of about 16 cm is over the sea surface. Drifters
transmit data in real time to a web portal called LD Manager where positions can be visualised in real time and data
downloaded in different formats. Each drifter was identified by a letter, after the prefix LC, for type of transmission or
sensors installed. So, *A* stands for a coastal GPRS drifter (LCA) while *E* for offshore satellite ones (LCE), *F* for offshore
drifters with the temperature sensor (LCF) and *H* for hybrid drifters (LCH). The latter transmits both by GPRS, when in
the GSM covered areas, and satellite when offshore. Furthermore, as most platforms are not equipped with additional
sensors, for the purposes of data uniformity we have not considered temperature data acquired with LCF drifters, but only
position data. Below the water, two different drogues, namely *Pila* and *Satis*, could be anchored through a swivel shackle.
The *Pila* was composed of two black joined plastic circles of 30 cm in diameter and used to follow the first layer of water,
while the *Satis* was an orange PVC sea-drogue floating anchor 50 cm long, similar to the drogues used for CLD and ODi
drifters, linked to the shackle through 3 mm polyester rope and positioned immediately below the drifter. Just in three
experiments in the northern Adriatic Sea, for specific project reasons, in 2018 the *Satis* drogue was positioned at 14 m
depth on drifters LCE00236 and at 20 m on LCE00234, and in 2019 at 14 m depth on drifter LCE00354. Data acquisition
frequency varied from 5 minutes to 12 hours between experiments, but also during a single track, because of several
situations or objectives like drifter deployment or recovery, distance from the coast, aim of the experiment. Usually for
Coastal Nomad drifters (LCA) we used frequencies of acquisition between 5 and 30 minutes while for Offshore Nomad
drifters from 15 minutes to 12 hours. Thanks to its ease of use, in drifter management or in data visualisation and
downloading, their use is still going on. Over the years they have been used for environmental and oceanographic studies
both at coastal and offshore scale but also for the validation of ocean forecasting and oil-spill systems in open ocean (SOS
Piattaforme project, http://www.seaforecast.cnr.it/sos-piattaforme) and coastal areas (Sicomar plus project;
http://www.seaforecast.cnr.it/sicomarplus) or ports (Geremia project, http://seaforecast.cnr.it/geremia). Experiments have
durations from a few hours to over 12 months with data covering most of the dataset presented here.
**3. Data processing method**
The drifter trajectories were submitted to a pre-processing immediately after the end of the experiment. Ancillary data
like temperature, battery level or drogue presence were not considered as these were not available for all platforms. From
each file, repeated positions or wrong date/time, generated by failure of the GPS receiver, were manually deleted. Data
from the CLD drifter, before the year 2000, displayed a large number of spikes as GPS was mainly for military use in that
period and a systematic position error (of 100 m) was intentionally added to the data. Over the years, the accuracy of the
positioning system has improved thanks to the increased availability of satellites and improved GPS receivers.
After the pre-processing, the drifter data of all the experiments were gathered in a unique excel file and sent to OGS to
be ingested and elaborated by the procedure schematically shown in (Fig. 3).

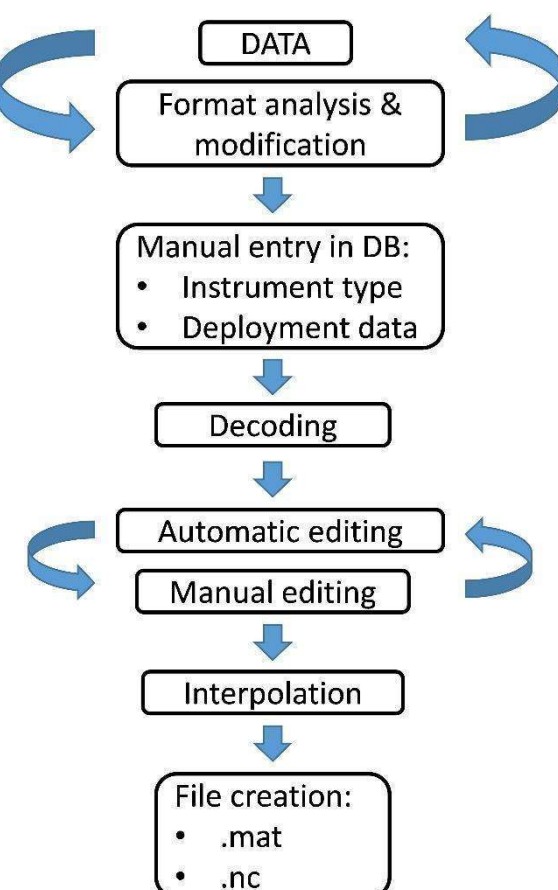

Figure 3. The processing procedure implemented at OGS from data acquisition (top) to file creation in Matlab/NetCDF
formats.

The OGS processing procedure is the result of more than 15 years of experience improving scripts and tests. It is capable
of handling over 80 different types of drifters and providing a common and therefore easily comparable set of files and
metadata (Gerin and Bussani, 2011; Menna et al., 2017). As a first step, the original excel file collecting all the tracks
was split into several text files corresponding to the data provided by the different drifters. These files may include data
from different experiments. Deployment and recovery information was retrieved from the original dataset and from the
experiment notes, and filled into a database management system based on the PostgreSQL free software
(https://www.postgresql.org/) at OGS. The database was then enriched with other important metadata such as the type
and characteristics of the instruments, the owner, and the principal investigator.
Ad-hoc decoding scripts were then implemented to associate the values contained in the files to the corresponding
parameters (i.e.: time, longitude, and latitude) and extract the data of the single experiment discarding repeated sets of
data. Exceeding spaces and spurious characters were removed to obtain data files compliant with the ASCII standard.
Decoded drifter data were then edited with the automatic procedure, through several QC tests, that replaced flagged time
and location data with *NaNs*. In particular, impossible drifter positions (longitude > 180 or < -180 and latitude > 90 or <
-90) and the positions on land were discarded. In the latter case, about 4000 polygons, extracted from the GEBCO 1-
minute resolution bathymetry data, which define the coordinates of all the coasts of the Mediterranean Sea, were used to
determine drifters not in the water. For experiments extremely near to the coastline, this last QC test was not carried out
to avoid the discarding of useful data. GPS data acquired before the beginning of the experiment and duplicated data due
to transmission repetitions were also flagged. In general, randomly, the GPS drifter data may display duplicated positions
acquired at different times. This was probably related to the buffer of the GPS module that does not correctly update the
position in its memory before transmitting the data. The automatic procedure considers this issue and marks this data as

incorrect. This procedure also evaluates the speed of the drifter. The first point (deployment position) was considered good and used as reference for the evaluation of the next point by computing the speed. If this speed exceeded 300 cm/s, the point was discarded and the evaluation is carried out on the further point, otherwise it was considered as a new reference and the procedure was iterated along all the available points. Additionally, a 4-degree polynomial fit was computed on a running window of 20 speed points, then speeds deviating from the fit by more than twice the total mean speed and twice the partial speed (computed considering only the points in the window) were not considered.

After the automatic editing procedure, some erroneous data still remained that required a visual check with a manual removal. In case of important temporal gaps or modification of the acquisition frequency during a Lagrangian experiment, the drifter trajectory was split into two segments and considered as two different deployments. New recovery/deployment information was included in the database and the automatic procedure relaunched. In the case of stranding, the automatic editing procedure discarded the data on land but is unable to recognise the moment when the drifter went ashore. The exact stranding time is defined by the operator through the visual analysis of the plotted drifter's trajectory.

Edited data were then interpolated at uniform intervals using a kriging optimum interpolation technique based on the correlation of the data (Hansen and Poulain, 1996). The technique adopts a structure function and weights that were previously estimated using the drifter data collected during other experiments in the Mediterranean Sea between 1986 and 2016, included in a dedicated dataset of over 2000 files (Menna et al., 2018b).

Drifter data with acquisition interval between a few minutes to 2 hours were interpolated at 1-hour intervals, then if more than 2 to 6 hours were interpolated at 3-h, and if more than 6 hours at 6-h intervals . The velocities were then calculated as central finite differences of the interpolated position. The velocity of the first and last point was set as 9999.

At the end of the whole procedure, the final dataset consists of 158 interpolated drifter's trajectories (Fig. 4) with at least two data points.

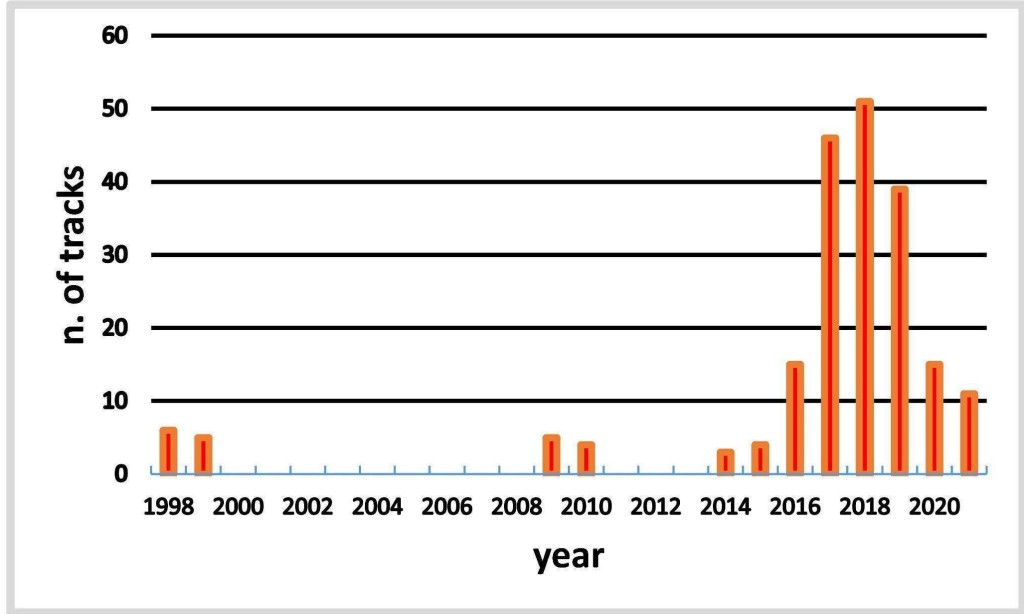

Figure 4. The histograms show number and distribution per year of drifter's trajectories between 1998 and 2021.

These tracks mainly cover the areas around Sardinia, the northern Tyrrhenian Sea (with the highest concentration of data for the whole period) and the Ligurian-Provençal basin.  A few drifters explored the Adriatic Sea, the Ionian Sea, and the Gulf of Lions (Fig. 5).

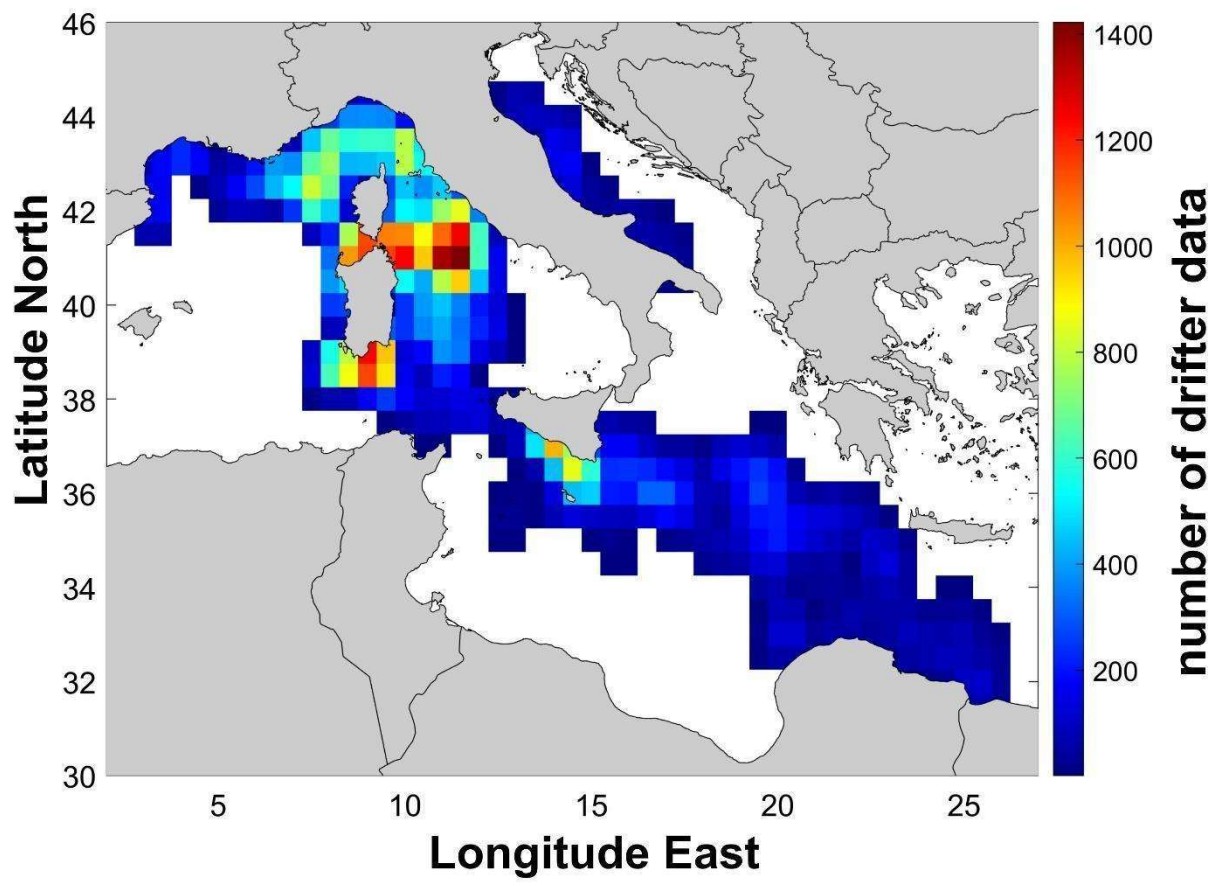

Figure 5. The distribution of drifters' data per pixel of half degree for the whole period 1998-2022. White pixels mean no data.

(Fig. 5) highlights the areas mainly of interest in several research projects that requested surface current experiments like the Bonifacio Strait, the northern Tyrrhenian Sea and the Sicily Strait often used for the validation of ocean numerical systems (Cucco et al., 2012; Ribotti et al., 2013).

**4. Data availability**

The dataset described is publicly available and free from the data repository in the SEANOE (SEA scieNtific Open data Edition) service at https://doi.org/10.17882/90537 (Ribotti et al., 2022) and at the SeaDataNet infrastructure at https://cdi.seadatanet.org/search/welcome.php?query=2610&query_code={9F00DF80-1881-42DD-9DF1-B9BD0282F2B0}. The presented dataset is composed of the interpolated data in NetCDF files which include time, latitude, longitude, zonal and meridional speed, and metadata. The dataset has been realised following international standards used for Lagrangian data and thought to be easily comparable with similar datasets. Variables definition and dimension follow two different formats: 1) the Copernicus Marine In Situ NetCDF format manual (https://archimer.ifremer.fr/doc/00488/59938/) that specifies the NetCDF file format of Copernicus Marine In Situ TAC used to distribute ocean In Situ data and metadata; 2) the NASA/NOAA-like NetCDF format manageable by Panoply, a data viewer developed by the NASA Goddard Institute for Space Studies (https://www.giss.nasa.gov/tools/panoply/). The dataset includes drifters' data with subsurface drogue (in the first meters) apart from a few experiments when the drogue was at 14 or 20 m depth (see par. 2.4). These experiments correspond to the files arib_LCE234 and brib_LCE234 (20 m), arib_LCE236 and arib_LCE354 (14 m) of the dataset.

**5. Discussion and conclusion**

Between mid-1998 and 2022, CNR collected drifters' data from more than a hundred experiments carried out in the Mediterranean in the framework of scientific and operational projects or international exercises at sea for preparedness and response activities to oil spill or SAR emergencies. Despite funding projects' objectives, experiments at sea were planned to use data also for different activities or scientific interests and/or needs like the validation of ocean circulation or oil spill models. So, as with any scientific measurement, there is always a duality between "fit for purpose", i.e., the projects that funded drifters and experiments, and "fit for use", i.e., the possibility of reusing the data for different objectives. This duality was facilitated by rechargeable drifters (most of those in the dataset) that, after recovering, could be used in further experiments and new data acquisitions.

Then, after the pre-processing of the data by the CNR in Oristano followed by the accurate elaboration by the OGS, all data in the dataset are comparable between them, even if realised with different drifters and in different years. Further, this dataset is also compliant and can be interfaced with the other drifter datasets produced by OGS in the Mediterranean and Black Sea which collect about 1700 drifter data starting from 1986 (Menna et al., 2017; Menna et al., 2018a; Menna et al., 2018b; Menna et al., 2019; Gerin et al., 2020), thus facilitating the use of a huge amount of drifter data available for scientific purposes in the Mediterranean basin (circulation, climate, etc).

Lastly, the dataset presented here collects 158 interpolated drifter tracks. Further data will be part of an additional dataset and comparable as it is processed according to the same criteria described in this paper.

**Author contribution**

AR led some projects with the use of drifters, all experiments, and the writing of the paper. AB finalised editing procedures described in the paper and collaborated on the paper writing. RG verified all data, realised the dataset, and collaborated on the paper writing. MM verified all processed data and collaborated on the paper writing. AS prepared all experiments and collaborated on the paper writing. RS and AC led some projects with the use of drifters and collaborated on the paper writing.

**Competing interests**

The authors declare that they have no conflict of interests.

**Acknowledgments**

We thank Mr. Mireno Borghini from CNR-ISMAR in La Spezia (Italy), the Italian Coast Guard, the Italian Navy, and captains and crews of all large and small vessels used to launch/recover drifters for their important support.

**Financial support**

The data used in this work have been collected in the framework of several national and European projects, i.e the Italian MATTM project SOS-BONIFACIO (prot. DPN-2009-0001027 of 20/01/2009), the Italian MIUR project PON TESSA (agreement PON01_02823), the Italian MIUR flagship project RITMARE (under the NRP 2011-2013, approved by the CIPE Resolution 2/2011 of 23.03.2011), the Italian MATTM project SOS-Piattaforme & Impatti offshore (Reg. Uff. U. 0000939.17-01-2017), 2014 - 2020 INTERREG V-A Italy - France (Maritime) project SICOMAR plus (IAS CNR Prot. 0001156/2018 of 12/12/2018).

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
