# Peer review of "A new released Mediterranean drifters' dataset"

_Earth System Science Data, 2022_

## Referee Comment (RC2)

REVISION OF ESSD-2022-344

The article entitled "A Mediterranean drifters dataset: 1998–2022" contains a series of files with trajectories of different kinds of drifters deployed between 1998 and 2021. Lagrangian drifters are a very useful tool for understanding the surface circulation of the ocean, as well as for determining mesoscale and submesoscale structures that are related to areas of accumulation of larvae or plastics, among others. However, in this work, the dataset needs to be revised in order to be used by other users in the scientific community. On one hand, title of the paper does not seem appropriate, since, as indicated in the manuscript, the first buoy deployment experiments were carried out in the years 1998-1999 but were not repeated until 10 years later, so this title is misleading. And on the other hand, some files (e.g. hrib_LCE230) are not related to Mediterranean tracks (they show latitudes of 68 – 70 ºN). The work needs a great deal of effort to improve its quality in order to be published.

**ABSTRACT**

Here, and along the manuscript, you say that the dataset belongs to 138 experiments but there are 204 tracks, could you explain why? trajectories of which drifters were divided?

**INTRODUCTION**

Are there previous Lagrangian experiments in the study area? Why is this dataset important?

**THE DRIFTERS**

Table 1 and Figure 1

As above mentioned, some files are not from the Mediterranean Sea. Please remove them from this dataset and article, or include them but change the study area and so on. Some of these drifters are: hrib_LCE230; mrib_LCE227; mrib_LCE229; prib_ LCE229; qrib_ LCE229; urib_ LCE229; vrib_ LCE229; (…).

In my opinion, in order to be able to draw conclusions about the Lagrangian motion, it is necessary to have several points of the buoy's motion, which allow for concluding how the circulation in an area is. In this work, many trajectories are presented with little data (less than 10), so I do not consider that they give information. I attach a table with all the available buoys and the number of data.

| Drifter name | number of data | Drifter name | number of data | Drifter name | number of data | Drifter name | number of data | Drifter name | number of data | Drifter name | number of data |
|---|---|---|---|---|---|---|---|---|---|---|---|
| aarib_LCA113 | 2 | arib_LCA122 | 33 | brib_CODE94657 | 4 | crib_LCE234 | 879 | jrib_LCA127 | 2 | vrib_LCE229 | 32 |
| aarib_LCA197 | 3 | arib_LCA123 | 126 | brib_CODE94658 | 62 | crib_LCE236 | 2799 | jrib_LCE227 | 29 | wrib_LCA116 | 2 |
| abrib_LCA197 | 4 | arib_LCA124 | 3 | brib_LCA075 | 4 | crib_LCE348 | 402 | jrib_LCE229 | 38 | wrib_LCE229 | 28 |
| acrib_LCA113 | 2 | arib_LCA125 | 26 | brib_LCA076 | 7 | crib_LCH353 | 363 | krib_CLD | 2 | xrib_LCE229 | 9 |
| aerib_LCA197 | 10 | arib_LCA126 | 94 | brib_LCA077 | 24 | drib_CLD | 4 | krib_LCA112 | 6 | yrib_LCA116 | 2 |
| afrib_LCA197 | 2 | arib_LCA127 | 2 | brib_LCA113 | 6 | drib_CODE94658 | 7 | krib_LCA113 | 2 | yrib_LCE229 | 41 |
| agrib_LCA197 | 3 | arib_LCA128 | 3 | brib_LCA116 | 7 | drib_LCA076 | 5 | krib_LCE227 | 14 | zrib_LCA113 | 2 |
| ahrib_LCA197 | 3 | arib_LCA132 | 82 | brib_LCA120 | 96 | drib_LCA127 | 31 | krib_LCE229 | 3 | zrib_LCA197 | 4 |
| airib_LCA116 | 2 | arib_LCE227 | 1231 | brib_LCA121 | 109 | drib_LCE227 | 5 | lrib_CLD | 5 | zrib_LCE229 | 4 |
| airib_LCA197 | 10 | arib_LCE228 | 30 | brib_LCA122 | 8 | drib_LCE229 | 4 | lrib_LCA113 | 3 | | |
| akrib_LCA116 | 2 | arib_LCE229 | 587 | brib_LCA124 | 18 | drib_LCE230 | 18 | lrib_LCE227 | 6 | | |
| akrib_LCA128 | 21 | arib_LCE230 | 482 | brib_LCA125 | 22 | drib_LCE234 | 71 | lrib_LCE229 | 98 | | |
| alrib_LCA113 | 7 | arib_LCE231 | 236 | brib_LCA128 | 5 | drib_LCE236 | 147 | mrib_CLD | 5 | | |
| alrib_LCA128 | 36 | arib_LCE232 | 271 | brib_LCE227 | 22 | drib_LCE348 | 51 | mrib_LCA112 | 31 | | |
| amrib_LCA113 | 21 | arib_LCE233 | 96 | brib_LCE229 | 17 | erib_LCE227 | 22 | mrib_LCE227 | 1010 | | |
| amrib_LCA128 | 25 | arib_LCE234 | 239 | brib_LCE230 | 29 | erib_LCE229 | 4 | mrib_LCE229 | 47 | | |
| aorib_LCA113 | 3 | arib_LCE235 | 64 | brib_LCE231 | 607 | erib_LCE230 | 12 | nrib_CLD | 5 | | |
| aprib_LCA113 | 27 | arib_LCE236 | 9 | brib_LCE232 | 300 | erib_LCE236 | 31 | nrib_LCA113 | 3 | | |
| arib_CLD | 2 | arib_LCE255 | 879 | brib_LCE233 | 72 | frib_LCA127 | 2 | orib_CLD | 5 | | |
| arib_CODE5748 | 140 | arib_LCE256 | 547 | brib_LCE234 | 197 | frib_LCA197 | 4 | orib_LCA127 | 3 | | |
| arib_CODE94656 | 2 | arib_LCE347 | 3950 | brib_LCE236 | 389 | frib_LCE227 | 7 | orib_LCE229 | 4 | | |
| arib_CODE94657 | 26 | arib_LCE348 | 175 | brib_LCE256 | 715 | frib_LCE229 | 14 | prib_LCA113 | 2 | | |
| arib_CODE94658 | 149 | arib_LCE349 | 100 | brib_LCE348 | 19 | frib_LCE230 | 67 | prib_LCA116 | 2 | | |
| arib_LCA075 | 67 | arib_LCE351 | 29 | brib_LCE349 | 45 | frib_LCE236 | 495 | prib_LCA127 | 3 | | |
| arib_LCA077 | 19 | arib_LCE354 | 7 | brib_LCE354 | 32 | grib_LCE227 | 77 | prib_LCE229 | 53 | | |
| arib_LCA078 | 89 | arib_LCE570 | 42 | brib_LCE570 | 196 | grib_LCE229 | 43 | qrib_LCE229 | 88 | | |
| arib_LCA079 | 67 | arib_LCE573 | 251 | brib_LCE573 | 298 | grib_LCE230 | 56 | rrib_LCA113 | 2 | | |
| arib_LCA109 | 101 | arib_LCE601 | 499 | brib_LCE579 | 5 | hrib_CLD | 4 | rrib_LCA127 | 3 | | |
| arib_LCA110 | 110 | arib_LCF425 | 438 | brib_LCE601 | 74 | hrib_LCE227 | 16 | rrib_LCE229 | 12 | | |
| arib_LCA111 | 119 | arib_LCH352 | 900 | brib_LCF425 | 84 | hrib_LCE229 | 154 | srib_LCA124 | 4 | | |
| arib_LCA112 | 4 | arib_ODi150 | 575 | brib_LCH353 | 6 | hrib_LCE230 | 285 | srib_LCA127 | 118 | | |
| arib_LCA113 | 3 | arib_ODi730 | 26 | brib_ODi730 | 18 | irib_CLD | 6 | srib_LCA197 | 3 | | |
| arib_LCA114 | 33 | aurib_LCA116 | 19 | crib_CODE94658 | 393 | irib_LCA127 | 2 | srib_LCE229 | 14 | | |
| arib_LCA115 | 82 | avrib_LCA116 | 45 | crib_LCA075 | 7 | irib_LCE227 | 10 | trib_LCA124 | 122 | | |
| arib_LCA116 | 3 | awrib_LCA116 | 53 | crib_LCA076 | 5 | irib_LCE229 | 49 | trib_LCA128 | 2 | | |
| arib_LCA118 | 29 | axrib_LCA116 | 33 | crib_LCE227 | 22 | jrib_CLD | 5 | trib_LCA197 | 3 | | |
| arib_LCA119 | 118 | azrib_LCA116 | 4 | crib_LCE229 | 25 | jrib_LCA112 | 2 | urib_LCA116 | 2 | | |
| arib_LCA120 | 21 | brib_CLD | 2 | crib_LCE230 | 8 | jrib_LCA113 | 3 | urib_LCE229 | 32 | | |
| arib_LCA121 | 71 | brib_CODE94656 | 7 | crib_LCE233 | 28 | jrib_LCA124 | 2 | vrib_LCA197 | 2 | | |

Therefore, I recommend removing the files with a such small number of locations.

Table 1 shows information about hybrid and temperature (LCH and LCF, respectively). Nevertheless, whe I downloaded the dataset I did not find information related to this kind of drifters. Besides this, please, remove temperature information in the NetCDF information (SST) because there is no data about that (as well as along the manuscript).

Following the data, you do not specify the units of zonal and meridional velocity in the files. I think you should explain how you calculate.

**Tracks 2009-100 2010: the CODE**

Line 102. Measurements of CODE 110 cm x 15 cm? These values are confusing attending to Figure 2B.

Line 115: Could you check it with your data? And not only with this kind of a drifter.

**Tracks 2015-2022: coastal and offshore Nomad drifters**

Line 156. Same here, in the dataset, there are no LCH and LCF files.

Line 157. Were the Southtek drifters drogue? Why is not shown in Figure 2?

Line 207. I do not understand: "A dedicated custom MATLAB tool…" What is this?

Line 213. "From the original 138…" Again, why? Which tracks?

Figure 4. Kind repetitive… you explain this in Table 1. There is a mistake in "th".

Line 225. Which figure?

---

## Author Comment (AC1)

To
Referee # 1
and Cc
Dr. Giuseppe M.R. Manzella
ESSD Chief Editor,
Dr. Alessio Rovere
ESSD topical editor
and
the ESSD Editorial Support Team

Oristano, 11/05/2023

Subject: reply to Referee's comments on "A Mediterranean drifters dataset: 1998–2022" by Alberto Ribotti, Antonio Bussani, Milena Menna, Andrea Satta, Roberto Sorgente, Andrea Cucco, and Riccardo Gerin, Earth Syst. Sci. Data Discuss., https://doi.org/10.5194/essd-2022-344

Dear Referee,

thank you very much for your valid suggestions. We have devoted our best efforts to improve the submitted manuscript, aided by your insightful comments. We conducted a point-by-point response to your comments and queries and the manuscript has been edited and corrected, accordingly. The details of these changes can be found in the ensuing point-by-point responses to each and every comment/suggestion.

Referee's queries are shown in *italics* to differentiate our replies introduced by a **REPLY:** in **bold**.

Best regards,
* * *
**Anonymous Referee #1**
**Referee comments**
**RC1**: 'Comment on essd-2022-344', Anonymous Referee #1, 16 Dec 2022

*This article is not fully comprehensible because of the English grammar, and as such, I deem it inappropriate for publication in its present stage. I would recommend a thorough editing and rewriting in order to improve the English.*

**REPLY:** Thank you for the comment. The article was completely revised by a native English-speaking colleague.

*I do not find the description of the processing methods to be extensive enough to understand the dataset. As an example, a kriging method is stated to be used to produce the estimated positions but nothing is said of the underlying structure functions that need to be first estimated to apply this*

*method. Another example is the quality control procedure of despiking: once again, no detail is given for this method (type of filter, threshold etc.).*

**REPLY:** The 3rd paragraph (Data processing method) has been re-written improving the description of the used processing methods and quality control procedures.

"Decoded drifter data were then edited with the automatic procedure, through several QC tests, that replaced flagged time and location data with NaNs. In particular, impossible drifter positions (longitude > 180 or <-180 and latitude >90 or <-90) and the positions on land were discarded. In the latter case, about 4000 polygons, extracted from the GEBCO 1-minute resolution bathymetry data, which define the coordinates of all the coasts of the Mediterranean Sea, were used to determine drifters not in the water. For experiments extremely near to the coastline, this last QC test was not carried out to avoid the discarding of useful data. GPS data acquired before the beginning of the experiment and duplicated data due to transmission repetitions were also flagged. In general, randomly, the GPS drifter data may display duplicated positions acquired at different times. This was probably related to the buffer of the GPS module that does not correctly update the position in its memory before transmitting the data. The automatic procedure considers this issue and marks this data as incorrect. This procedure also evaluates the speed of the drifter. The first point (deployment position) was considered good and used as reference for the evaluation of the next point by computing the speed. If this speed exceeded 300 cm/s, the point was discarded and the evaluation is carried out on the further point, otherwise it was considered as a new reference and the procedure was iterated along all the available points. Additionally, a 4-degree polynomial fit was computed on a running window of 20 speed points, then speeds deviating from the fit by more than twice the total mean speed and twice the partial speed (computed considering only the points in the window) were not considered.

After the automatic editing procedure, some erroneous data still remained that required a visual check with a manual removal. In case of important temporal gaps or modification of the acquisition frequency during a Lagrangian experiment, the drifter trajectory was split into two segments and considered as two different deployments. New recovery/deployment information was included in the database and the automatic procedure relaunched. In the case of stranding, the automatic editing procedure discarded the data on land but is unable to recognise the moment when the drifter went ashore. The exact stranding time is defined by the operator through the visual analysis of the plotted drifter's track.

Edited data were then interpolated at uniform intervals using a kriging optimum interpolation technique based on the correlation of the data (Hansen and Poulain, 1996). The technique adopts a structure function and weights that were previously estimated using the drifter data collected during other experiments in the Mediterranean Sea between 1986 and 2016, included in the db_med24_nc_1986_2016 dataset (about 2000 files; Menna et al., 2017).

Drifter data with acquisition frequency between a few minutes to 2 hours were interpolated at 1-hour intervals, while those with acquisition frequency till or more than 6 hours were interpolated at 3-h and 6-h intervals, respectively. The velocities were then calculated as finite differences of the interpolated position."

*The dataset website (https://doi.org/10.17882/90537) indicates that 366 trajectories (tracks) are available yet the article mentions 204? After downloading all the files, the number of track appears to be indeed 204, one per file.*

**REPLY:** These tracks were discarded from the dataset and then drifters' numbers were recalculated accordingly.

*These files do not follow a traditional data format: every single variable in these files (u,v, Lat, Lon, etc.) has its own dimension with the name of the variable. In other words, the variable "u" has dimension "u", which is odd. This does not suggest that these variables are contemporaneous or constitute time series along a common dimension ("obs" as an example). Moreover, because some variables exhibit missing values, a common software like Panoply is unable to plot time series for which missing values are present (because the dimension for that variable has missing values!). My suggestion is to reformat and recreate these files so that the variables have a common dimension (such as "obs"). There are template available for trajectory files, see as an example the one from NOAA NCEI (https://www.ncei.noaa.gov/netcdf-templates).*

**REPLY:** The dataset has been realised following international standards used for Lagrangian data and thought to be easily comparable with similar datasets. Variables definition and dimension follow the Copernicus Marine In Situ NetCDF format manual (https://archimer.ifremer.fr/doc/00488/59938/) that specifies the NetCDF file format of Copernicus Marine In Situ TAC used to distribute ocean In Situ data and metadata. Moreover the dataset was ingested in SeadataNet following international standards and is also available at the address https://cdi.seadatanet.org/search/welcome.php?query=2610&query_code={9F00DF80-1881-42DD-9DF1-B9BD0282F2B0}. Such a link was also added in the text.
We agree with the reviewer that those variables which are all NaN are unnecessary and may be inconvenient to handle, so we have removed them from this dataset.

*Moreover, some files have only two data points for each variables, and in the particular example of aarib_LCA113.nc, no valid value at all. What is the point of this set of data? This shows inadequate curation or automatic processing and editing of the data.*

**REPLY:** The whole dataset was checked again. We have chosen to generate a dataset with three different interpolation frequencies (1-h, 3-h and 6-h) and to include all available trajectories with at least two measurements, then leaving the user the choice whether to use them or not.
The 1-hour interpolation, used for short experiments (a bit more than an hour), provides a few points only. These data may seem insignificant but, when put together with data from other drifters in the same area, they can constitute an important source of information. For example, they can contribute by describing the surface circulation in the basin by pseudo-Eulerian statistics, as described by Poulain (2001; https://doi.org/10.1016/S0924-7963(01)00007-0).

---

## Author Comment (AC2)

To

Referee #2

and Cc

Dr. Giuseppe M.R. Manzella

ESSD Chief Editor,

Dr. Alessio Rovere

ESSD topical editor

and

the ESSD Editorial Support Team

Oristano, 11/05/2023

Subject: reply to Referee's comments on "A Mediterranean drifters dataset: 1998–2022" by Alberto Ribotti, Antonio Bussani, Milena Menna, Andrea Satta, Roberto Sorgente, Andrea Cucco, and Riccardo Gerin, Earth Syst. Sci. Data Discuss., https://doi.org/10.5194/essd-2022-344

Dear Referee,

thank you very much for your valid suggestions. We have devoted our best efforts to improve the submitted manuscript, aided by your insightful comments. We conducted a point-by-point response to your comments and queries and the manuscript has been edited and corrected, accordingly. The details of these changes can be found in the ensuing point-by-point responses to each and every comment/suggestion.

Referee's queries are shown in *italics* to differentiate our replies introduced by a **REPLY:** in **bold**.

Best regards,
* * *
**Anonymous Referee #2**
**Referee comments**
**RC2**: 'Comment on essd-2022-344', Anonymous Referee #2, 04 Apr 2023

*The article entitled "A Mediterranean drifters dataset: 1998–2022" contains a series of files with trajectories of different kinds of drifters deployed between 1998 and 2021. Lagrangian drifters are a very useful tool for understanding the surface circulation of the ocean, as well as for determining mesoscale and submesoscale structures that are related to areas of accumulation of larvae or plastics, among others. However, in this work, the dataset needs to be revised in order to be used by other users in the scientific community.*

*On one hand, title of the paper does not seem appropriate, since, as indicated in the manuscript, the first buoy deployment experiments were carried out in the years 1998-1999 but were not repeated until 10 years later, so this title is misleading.*

**REPLY:** We agree with the referee. We have changed the title in "A new released Mediterranean drifters' dataset"

*And on the other hand, some files (e.g. hrib_LCE230) are not related to Mediterranean tracks (they show latitudes of 68 – 70 ºN). The work needs a great deal of effort to improve its quality in order to be published.*

**REPLY:** We carefully checked the tracks and eliminated those outside the Mediterranean. Numbers of drifters' trajectories were recomputed accordingly.

**ABSTRACT**

*Here, and along the manuscript, you say that the dataset belongs to 138 experiments but there are 204 tracks, could you explain why? trajectories of which drifters were divided?*

**REPLY:** We have simplified the text just mentioning the experiments without numbers that confuse the reader.

**INTRODUCTION**

*Are there previous Lagrangian experiments in the study area? Why is this dataset important?*

**REPLY:** There have been a lot of Lagrangian experiments in the Med area which are published by OGS in the Mediterranean and Black Sea dataset (approximately 1700 drifter data since 1986): see for example 10.6092/B40CD642-9555-44FA-8B91-3CD88B6C225B, 10.6092/7a8499bc-c5ee-472c-b8b5-03523d1e73e9.
This dataset is important because it is composed of new data generated following the same procedures as the above-mentioned OGS datasets. This permits researchers to integrate and compare these data with previous released ones for scientific Mediterranean studies on circulation, climate, etc. We mentioned this in the Introduction leaving understood its importance, then highlighted in the Discussion and conclusions.
In the Introduction (page 2, lines 68-70 of the file with revisions) we wrote: "Recently, the OGS in Trieste has re-elaborated all drifters' experiments following standard and state-of-the-art procedures (editing and interpolation) already adopted for previously released Lagrangian datasets, then creating a new one freely available online."
In the Discussion and conclusion (page 12, lines 279-283 of the file with revisions): "Further, this dataset is also compliant and can be interfaced with the other drifter datasets produced by OGS in the Mediterranean and Black Sea which collect about 1700 drifter data starting from 1986 (Menna et al., 2017; Menna et al., 2018a; Menna et al., 2018b; Menna et al., 2019; Gerin et al., 2020), thus facilitating the use of a huge amount of drifter data available for scientific purposes in the Mediterranean basin (circulation, climate, etc)."

**THE DRIFTERS**

*Table 1 and Figure 1: As above mentioned, some files are not from the Mediterranean Sea. Please remove them from this dataset and article, or include them but change the study area and so on. Some of these drifters are: hrib_LCE230; mrib_LCE227; mrib_LCE229; prib_LCE229; qrib_ LCE229; urib_ LCE229; vrib_ LCE229; (...).*

**REPLY:** We have deleted tracks outside the Mediterranean.

*In my opinion, in order to be able to draw conclusions about the Lagrangian motion, it is necessary to have several points of the buoy's motion, which allow for concluding how the circulation in an area is. In this work, many trajectories are presented with little data (less than 10), so I do not consider that they give information. I attach a table with all the available buoys and the number of data. Therefore, I recommend removing the files with a such small number of locations.*

**REPLY:** We are conscious of this but we think that also the very short tracks can be useful to describe the surface circulation in the basin by, for example, pseudo-Eulerian statistics as described by Poulain (2001; https://doi.org/10.1016/S0924-7963(01)00007-0). We left the user the choice to use them or not.

*Table 1 shows information about hybrid and temperature (LCH and LCF, respectively). Nevertheless, when I downloaded the dataset I did not find information related to this kind of drifters. Besides this, please, remove temperature information in the NetCDF information (SST) because there is no data about that (as well as along the manuscript).*

**REPLY:** We have eliminated this information from the dataset. We have explained the missing of these data with the sentence "Ancillary data like temperature, battery level or drogue presence were not considered as not available for all platforms" in the first lines of the 3rd paragraph.

*Following the data, you do not specify the units of zonal and meridional velocity in the files. I think you should explain how you calculate.*

**REPLY:** We have added the units of the zonal and meridional velocities in the metadata of the dataset. The explanation of the velocity computation is provided at the end of paragraph 3 with the following sentence "The velocities were then calculated as finite differences of the interpolated positions."

**Tracks 2009-100 2010: the CODE**

*Line 102. Measurements of CODE 110 cm x 15 cm? These values are confusing attending to Figure 2B.*

**REPLY:** The mentioned dimensions were indicating just the housing of the drifter, not well visible in the figure. We have decided to update the sentence, then considering the dimension of the whole instrument, sails included.

*Line 115: Could you check it with your data? And not only with this kind of a drifter.*

**REPLY:** Different types of drifters acquired with different sampling intervals from a few minutes to several hours but the dataset includes just processed interpolated data and not raw ones. As stated in the text, drifter data with an acquisition frequency between a few minutes to 2 hours were interpolated at 1-hour intervals, while those with acquisition frequency till or more than 6 hours were interpolated at 3-h and 6-h intervals, respectively.

**Tracks 2015-2022: coastal and offshore Nomad drifters**

*Line 156. Same here, in the dataset, there are no LCH and LCF files.*

**REPLY:** We carefully checked and there are just a few LCH and LCF files in the dataset. They are visible at columns 2, 3 and 4 of the table in the Referee's comments.

*Line 157. Were the Southtek drifters drogue? Why is not shown in Figure 2?*

**REPLY:** All types of drifters, Southtek ones included, were equipped with a drogue. This information was missing at paragraph 2.1 on CLD drifter where we added the sentence "A sail (0.5 m length and diameter) was attached below the drifter to enhance the drag below the water surface.".
Drogues figures and their dimensions have been added to Figure 2.

*Line 207. I do not understand: "A dedicated custom MATLAB tool..." What is this?*

**REPLY:** We are sorry for the misunderstanding. We meant script and not tool. Anyway we modified the sentence to better detail the process, as follows: "The exact stranding time is defined by the operator through the visual analysis of the plotted drifter's track."

*Line 213. "From the original 138..." Again, why? Which tracks?*

**REPLY:** In order to avoid any confusion, we have rephrased the sentence and simplified it as follows: "At the end of the whole procedure, the final dataset consists of 158 interpolated drifter tracks (Figure 4) with at least two data points.".

*Figure 4. Kind repetitive... you explain this in Table 1. There is a mistake in "th".*

**REPLY:** We have re-plotted the figure by considering only the tracks.

*Line 225. Which figure?*

**REPLY:** We have modified the sentence by indicating the number of the figure mentioned in the text (Figure 5).

---

## Author Comment (AC3)

To

CC1

and Cc

Dr. Giuseppe M.R. Manzella

ESSD Chief Editor,

Dr. Alessio Rovere

ESSD topical editor

and

the ESSD Editorial Support Team

Oristano, 11/05/2023

Subject: reply to Referee's comments on "A Mediterranean drifters dataset: 1998–2022" by Alberto Ribotti, Antonio Bussani, Milena Menna, Andrea Satta, Roberto Sorgente, Andrea Cucco, and Riccardo Gerin, Earth Syst. Sci. Data Discuss., https://doi.org/10.5194/essd-2022-344

**CC1: 'Comment on essd-2022-344', Adam Gauci, 10 Jan 2023**

*This work presents various drifter experiments that were carried out between 1998 and 2022. In Section 2 of the paper, the authors put forward the characteristics and specifications of the four types of drifters used. Very good and relevant comparisons between the instrument dimensions, battery duration, data transmission type, and data formats, are made. Clearly, the authors have extensive knowledge and experience in using and deploying such equipment. The points mentioned are useful to anyone that plans to carry out similar experiments in the same region or elsewhere. In Section 3, details about the data quality control procedures are provided. This has also been carried out meticulously and follows the current state-of-the-art methods. Details of how the data is saved and what information is stored are also mentioned. The 138 trajectories that are discussed are also made freely available. This dataset is useful to several users working with data assimilation or HF radar validation. Well done to the authors. Excellent work!*

REPLY: We thank Dr. Adam Gauci for his assessment of our work. We hope that the dataset will prove useful and that the manuscript in its final form, after careful revision, will be of more interest.

Best regards,

---

## Referee Report (RR1)

REVISION OF ESSD-2022-344-R1

In this version of the manuscript, the authors have taken into account all the changes suggested in the previous revision. The title of the paper has been changed, and the data uploaded to the SEANOE platform has been revised.

Here are some comments:

**INTRODUCTION**

Could you add some information in Fig.1 about the locations that you mention in the text? It's just to have an idea about the available drifters in every place.

**THE DRIFTERS**

**Tracks 1998-1999**

Line 95: It is necessary to add information about the location of the Company? You do not give such information to others.

**Tracks 2015-2022: coastal and offshore Nomad drifters**

Line 169: you mention that LCFs are provided with a temperature sensor, but you do not give the temperature information in the NetCF. Why? If you do not put this information I'll mention it in the text because, in my opinion, it's a bit confusing for the readers. They could think that they could use temperature information for those drifters.

Line 182: you could mention other studies where Southtek drifters have also been employed:

Sala I., Bolado-Penagos M., Bartual A., Bruno M., García C.M., López-Urrutia A., González-García C. and F. Echevarría (2022). A Lagrangian approach to the Atlantic Jet entering the Mediterranean Sea: Physical and biogeochemical characterization. Journal of Marine Systems. 2022, 226, doi: 10635210.1016/j.jmarsys.2021.103652

Bolado-Penagos M., González C. J., Chioua J., Sala I., Gomiz-Pascual J.J., Vázquez A. and M. Bruno (2020). Submesoscale processes in the coastal margins of the Strait of Gibraltar. The Trafalgar – Alboran connection. Progress in Oceanography, 2020, 118, 102219 DOI: 10.1016/j.pocean.2019.102219

**THE DRIFTERS**

Line 199: you have mentioned the other figures in the text as: "Fig.", so please, write Fig. 3 similarly.

Line 262: same comment for Fig. 5.

**DATA AVAILABILITY**

Line 274: add "s" to "in the first meters".

---

## Author Response (AR2)

To
Dr. Alessio Rovere
ESSD topical editor,
Referees #1-2
and Cc
Dr. Giuseppe M.R. Manzella
ESSD Chief Editor,
and
the ESSD Editorial Support Team

Oristano, 01/09/2023

Subject: reply to Referee's comments on "A new released Mediterranean drifters' dataset" by Alberto Ribotti, Antonio Bussani, Milena Menna, Andrea Satta, Roberto Sorgente, Andrea Cucco, and Riccardo Gerin, Earth Syst. Sci. Data Discuss., https://doi.org/10.5194/essd-2022-344

Dear Topic Editor,

Thank you very much for your hard work in choosing and following referees and your valid suggestions. Below we answer to your final comment on the dataset's format:

**COMMENT:** … However, one reviewer has a very important remark regarding your NETcdf file format, which apparently is not up to the Copernicus standard. I strongly encourage you to answer to all new comments, with particular regards to this one, so we can proceed with your MS.

**REPLY:** as answered to the Referee's 2 comment probably the Copernicus format is not the best for describing Lagrangian data; nevertheless, it allows Lagrangian data to be included. For consistency with other datasets developed and published in the past, we prefer to keep the Copernicus dataset as well. The metadata of the Copernicus-compliant files were updated to Copernicus format 1.6 and the files were successfully scanned with the Copernicus format checker (version 1.16). Furthermore the dataset was completely revised following the example given by the referee and the metadata was updated. The new NetCDF files were successfully tested with the Panoply programme that was indicated by the same referee in his first review.

Best regards,

Dear Referees,

thank you very much for your valid suggestions. We have devoted our best efforts to improve the submitted manuscript, aided by your insightful comments. We conducted a point-by-point response to your comments and queries and the manuscript has been edited and corrected, accordingly. The details of these changes can be found in the ensuing point-by-point responses to each and every comment/suggestion.

Referee's queries are shown in *italics* to differentiate our replies introduced by a **REPLY:** in **bold**.

**Anonymous Referee #1**
**Referee comments**
**RC2**: 'Comment on essd-2022-344', Anonymous Referee #1, 31 July 2023

*In this version of the manuscript, the authors have taken into account all the changes suggested in the previous revision. The title of the paper has been changed, and the data uploaded to the SEANOE platform has been revised.*

*Here are some comments:*

*INTRODUCTION Could you add some information in Fig.1 about the locations that you mention in the text? It's just to have an idea about the available drifters in every place.*

**REPLY:** We have re-plotted the Fig.1 adding most of the locations mentioned in the text. It was not always possible due to the presence of drifters' tracks

*THE DRIFTERS*

*Tracks 1998-1999 Line 95: It is necessary to add information about the location of the Company? You do not give such information to others.*

**REPLY:** the referee is right. The location has been deleted.

*Tracks 2015-2022: coastal and offshore Nomad drifters*

*Line 169: you mention that LCFs are provided with a temperature sensor, but you do not give the temperature information in the NetCF. Why? If you do not put this information I'll mention it in the text because, in my opinion, it's a bit confusing for the readers. They could think that they could use temperature information for those drifters.*

**REPLY:** The referee is right and so we added the following sentence after having mentioned LCF drifters at line 169: "Furthermore, as most platforms are not equipped with additional sensors, for the purposes of data uniformity we have not considered temperature data acquired with LCF drifters, but only position data." Another sentence is however also taken up in the paragraph on data processing and considers temperature data from all drifters described in the previous paragraphs.

*Line 182: you could mention other studies where Southtek drifters have also been employed:*

*Sala I., Bolado-Penagos M., Bartual A., Bruno M., García C.M., López-Urrutia A., González-García C. and F. Echevarría (2022). A Lagrangian approach to the Atlantic Jet entering the Mediterranean Sea: Physical and biogeochemical characterization. Journal of Marine Systems. 2022, 226, doi: 10635210.1016/j.jmarsys.2021.103652*

*Bolado-Penagos M., González C. J., Chioua J., Sala I., Gomiz-Pascual J.J., Vázquez A. and M. Bruno (2020). Submesoscale processes in the coastal margins of the Strait of Gibraltar. The Trafalgar – Alboran connection. Progress in Oceanography, 2020, 118, 102219 DOI: 10.1016/j.pocean.2019.102219*

**REPLY:** We have used these two references in the first sentence of the paragraph on Southtek's drifters

*Line 199: you have mentioned the other figures in the text as: "Fig.", so please, write Fig. 3 similarly.*

**REPLY:** We have modified it accordingly.

*Line 262: same comment for Fig. 5.*

**REPLY:** We have modified it accordingly.

*DATA AVAILABILITY*

*Line 274: add "s" to "in the first meters".*

**REPLY:** We have added it accordingly.

**Anonymous Referee #2**
**Referee comments**
**RC2**: 'Comment on essd-2022-344', Anonymous Referee #2, 01 July 2023

*"A new released Mediterranean drifters' dataset": That is not a very good title as it is doomed to become obsolete. You should consider something more general such as "A dataset of drifters in the Mediterranean Sea".*

**REPLY:** We agree with the referee. We have changed the title in "A Mediterranean drifters' dataset"

*The manuscript is greatly improved from the original submission. It still contains some odd grammatical phrases and I provide below some indications on how to correct those.*

*My on-going and main concern is about the format of the NetCDF files. These still contains variables with dimensions of the same name. As an example, the variable "Lon" has dimension "Lon", the variable "u" has dimension "u" etc. Because your drifters do not share a common uniform time dimension, there is only one dimension/coordinate per drifter and per file and that is something that could be called "obs" or "observation" or again "index". See as an example the individual NetCDF files of the NOAA Global Drifter Program. You state that you follow a Copernicus format standard but this one clearly does not apply to Lagrangian data. I do not approve of the distribution of this Lagrangian dataset with such a format. The editor may disagree with me.*

**REPLY:** We agree with the referee. We have changed the title in "A Mediterranean drifters' dataset"

*l11: realised -> conducted*

**REPLY:** We have corrected accordingly.

*l48: were interrupted to restart : were interrupted but restarted?*

**REPLY:** The referee is right. We have corrected accordingly.

*l61: (and after) enterprise -> company? manufacturer?*

**REPLY:** We have changed with "company".

*l62: What is GPRS?*

**REPLY:** We have added the following sentence: "GPRS (or General Packet Radio Service) was the first cellular system specifically designed for packet-switched, medium-speed data transfer over a cellular network, so it can only be used in coastal areas covered by a cellular network and is less expensive than GPS."

*l68: re-elaborated? re-processed?*

**REPLY:** We have corrected in "re-processed"

*l87: realised? -> built, manufactured?*

**REPLY:** We have corrected in "produced"

*l108: triangulation? -> positioning*

**REPLY:** We have cancelled "triangulation" that can lead to misinterpretation of the difference in positioning between ARGOS and GPS satellite systems. We thank the referee for this comment.

*l119: strictly linked with the presence of satellites: What does this mean?*

**REPLY:** ARGOS uses a Doppler system to calculate the position of an object on Earth. The object sends tags to the satellites passing over its area that turn this information to a data centre on Earth. This centre calculates the position of the object through the variation of the position of the satellite, its speed and distance from the Earth, and other parameters then transmitting the calculated position directly to the owner of the object. At least 4 satellites are necessary to have an acceptable error in the calculation of the position.

We preferred to avoid such a long description that can confuse the reader. Since there is already a short sentence in the previous lines on this point (l114-l116), we preferred to delete the sentence correctly underlined by the referee.

*l142: during experiments: that is obvious and I would remove.*

**REPLY:** The referee is right. We have cancelled it.

*l152: made in plastic -> made of plastic*

**REPLY:** Corrected accordingly.

*l224: "db_med24_nc_1986_2016": what is that?*

**REPLY:** The referee is right. We have modified the sentence (and reference) as follows: "included in a dedicated dataset of over 2000 files (Menna et al., 2018b)."

*l226: till? What do you mean?*

**REPLY:** The referee is right. We have re-written the sentence as follows: "Drifter data with acquisition frequency between a few minutes to 2 hours were interpolated at 1-hour intervals, then if more than 2 to 6 hours were interpolated at 3-h, and if more than 6 hours at 6-h intervals."

*l225-226: You keep mentioning acquisition frequency but really you are describing acquisition period or interval.*

**REPLY:** The referee is right. I have changed "frequency" with "interval"

*l227: central, forward, or backward finite differences? What do you do for end points?*

**REPLY:** The velocities were calculated considering the central finite differences. We added this information in the text and indicated that the velocity of the first and last point was set as 9999.

*l248: See my general comment.*

**REPLY:** The dataset was completely revised following the example given by the referee and the metadata was updated. Now the variables Time, Lon, Lat, u and v have a single variable as dimension, which has been named 'obs'. The new NetCDF files were successfully tested with the Panoply programme that was indicated by this referee in his first review.

Probably the Copernicus format is not the best for describing Lagrangian data; nevertheless, it allows Lagrangian data to be included. For consistency with other datasets developed and published in the past, we prefer to keep the Copernicus dataset as well. The metadata of the Copernicus-compliant files were updated to Copernicus format 1.6 and the files were successfully scanned with the Copernicus format checker (version 1.16).

The DOI on SEANOE will consequently have two distinct datasets (one according to the format useful for Panoply and another according to the Copernicus standard). The end user will be then free to choose the dataset that suits him best.

The text of the paper has been modified in several places to indicate this bipartition of datasets.

*l269-270: Will you be implementing a versioning system?*

**REPLY:** This was our initial idea but during this revision we verified that it is better to create further datasets. We have changed the final sentences as follows: "Lastly, the dataset presented here collects 158 interpolated drifter tracks. Further data will be part of an additional dataset and comparable, as they will be processed according to the same criteria described in this paper."